# Development and Validation of a Zone Fire Model Embedding Multi-Fuel Combustion

Bernard Porterie [1,2,3,*], Yannick Pizzo [1,2,3], Maxime Mense [4], Nicolas Sardoy [4], Julien Louiche [4], Nina Dizet [1,2], Timothé Porterie [3] and Priscilla Pouschat [3]

[1] University Institute of Industrial Thermal Systems (IUSTI), Aix Marseille University, National Centre for Scientific Research (CNRS), 13013 Marseille, France; yannick.pizzo@univ-amu.fr (Y.P.); nina.dizet@gmail.com (N.D.)

[2] Scientific Research and Numerical Simulation (RS2N), 83640 Saint-Zacharie, France

[3] Innovation and Development (INNODEV), 13013 Marseille, France; timothep@protonmail.com (T.P.); priscillapouschat@hotmail.com (P.P.)

[4] Délégation Générale de l'Armement–Techniques Navales, 83050 Toulon, France; maxime.mense@intradef.gouv.fr (M.M.); nicolas.sardoy@intradef.gouv.fr (N.S.); julien.louiche@intradef.gouv.fr (J.L.)

[*] Correspondence: bernard.porterie@univ-amu.fr; Tel.: +33-667934978

**Featured Application: This study provides confidence in the application of the zone model to describe fire growth and smoke transport in compartments where complex and multiple fuels are involved.**

**Abstract:** This paper presents the development and validation of a two-zone model to predict fire development in a compartment. The model includes the effects of the ceiling jet on the convective heat transfer to enclosure walls and, unlike existing models, a new concept of surrogate fuel molecule (SFM) to model multi-fuel combustion, and a momentum equation to accurately track the displacement of the smoke layer interface over time. The paper presents a series of full-scale fire experiments conducted in the IUSTI fire laboratory, involving different combinations of solid and liquid fuels, and varying the compartment confinement level. The model results are compared to the experimental data. It was found that for all fire scenarios, the experimental trends are well reproduced by the model. The SFM concept predicts oxygen and carbon dioxide concentrations in the extracted smoke to within a few percent of the measurements, which is a good agreement considering the sensitivity of the model to chemical formulas and combustion properties of fuels. Comparison with other measurements, namely average gas and wall temperatures, is also good. For the large fires reported in this study, the impact of the ceiling jet leads to a slight underestimation of wall temperatures, while the model gives conservative estimates for small fires.

**Keywords:** fire safety; two-zone model; full-scale fire experiments; multi-fuel combustion; surrogate fuel molecule; validation

## 1. Introduction

Because they represent a good compromise between speed and accuracy, fire zone models are always widely used by the research community and fire safety engineers. Compared to more sophisticated models, such as CFD (computational fluid dynamics) models, the computational run times of zone models are significantly shorter, on the order of a minute, or even less. In addition, they require little memory and few data, which makes them still particularly attractive.

Zone models may be classified based on the number of zones in each compartment: one-zone, two-zone, and multi-zone models [1,2]. One-zone models are widely used in the analysis of post-flashover fires, as well as smoke transport in the compartments separate

from the fire room (e.g., COMPF2 [3] or OZone [4]). In a two-zone model, the compartment is divided into two uniform well-mixed zones: a hot upper smoke layer and a cold lower layer, separated by a planar interface. They solve conservation equations between the two zones and use empirical relationships to describe phenomena such as fire plume, flame height, air entrainment, ceiling jet, or flows at vents. The multi-zone model is an extension of the two-zone model as the room volume is divided into an arbitrary number of horizontal layers, in which the physical properties (e.g., temperature or gas concentrations) are assumed to be uniform. They were developed to better predict the vertical distributions of temperature and gas concentrations in the fire room. Due to the assumptions on which they are based, two- and multi-zone models exhibit drawbacks or have limitations when applied to spaces of complex geometry or large size, but as mentioned before they give satisfactory results at lower cost and, because of their capabilities, they can be used for pre- and post-flashover fire modeling. Examples of such models developed since the 2000's are CFAST [5], MAGIC [6], BRI$_{2002}$ [7], B-RISK [8,9], and BRANZFIRE [10] (among others).

In existing two-zone models, the pressure is assumed to be uniform within a compartment and the momentum of the smoke interface is ignored, which avoids the time step imposed by acoustic waves (Courant condition). However, for under-ventilated fire scenarios, this strategy can lead to total room involvement by the smoke layer, which is not observed experimentally [11].

Some zone models, such as CFAST, include the ability to track multiples fires in the compartment, but these fires are treated as totally separate entities, with no interaction of the plumes or radiative exchanges between fires [5]. This approach allows multiple fires to be handled when they are far enough apart.

In an attempt to address these issues, a two-zone computational model is developed and validated to determine the fire environment inside a compartment where different fuels, close to each other, are burning simultaneously. The model includes a momentum equation to improve the accuracy of tracking the smoke layer interface and a new concept of surrogate fuel molecule to mimic multiple fuel combustion, as well as the effects of the ceiling jet on the convective heat transfer at the enclosure walls.

This paper is organized as follows. Section 2 deals with the mathematical basis of the model. Section 3 describes the experimental setup and the ten multiple fire scenarios that have been conducted in the IUSTI facility, varying the fuels involved and the confinement level of the enclosure. Section 4 provides a comparative analysis between the model results and the experimental data to evaluate the performance of the model developed.

## 2. The Model

The two-zone model proposed is based on concepts similar to those used in CFAST [5]. However, it uses a new form of the governing equations, combustion, and heat transfer sub-models which are detailed in the following sections.

### 2.1. Model Assumptions

The model is based on the following key assumptions:

- The compartment is assumed to be a rectangular parallelepiped with size $V_{room} = x_{room}y_{room}z_{room}$. It is divided into two zones separated by an interface: a hot zone containing the combustion products, and for certain ventilation conditions, the excess air and unburned fuel, and a cold zone containing fresh air (Figure 1). This assumes that the two zones coexist permanently and that the hot smoke layer is well stratified. At the start of the simulation, the layers are initialized to ambient conditions and the upper layer volume is set to an arbitrary small value of the compartment volume (here, 0.001).
- Temperature, density, pressure, and species concentrations are assumed to be uniform in each zone. However, unlike other zone models such as CFAST, the pressure is different from one zone to the other. Therefore, in the present model, the set of governing equations includes not only the conservation equations of mass and energy,

but also a momentum equation that governs the displacement of the smoke interface as a function of the pressure difference between the two zones. Solving this momentum equation leads to a reduction in the time step, imposed by the propagation of acoustic waves (Courant-Friedrichs-Levy or CFL condition), but allows us to follow more accurately the displacement of the interface.

- To model radiation heat transfer in the compartment, the ten-surface model [12] is applied. These ten surfaces, hereafter called wall segments, are the ceiling, four upper walls (i.e., located above the smoke interface), four lower walls (i.e., located below the smoke interface), and the floor (Figure 2). Each wall segment is assumed to be at a uniform temperature. The fire is assumed to radiate uniformly in all directions from a point source at the center of the flame, located at one third of the flame height, given off a fraction $\chi_r$ of the total energy release rate to thermal radiation. The radiation emitted from a wall surface, a gas layer, and the fire is assumed to be grey and diffuse. Radiation transfer through vent openings, doors, etc., is neglected.
- The transient pyrolysis rate for each fuel involved by the fire is prescribed by the user (here, it is deduced from the experimentally measured mass loss rate history), but it may be constrained by the availability of oxygen in the compartment.
- The specific heats at constant volume and pressure, $c_v$ and $c_p$, are assumed to be constant. They are related to the individual gas constant $R$ and the ratio of specific heats $\gamma$ by: $\gamma = c_p/c_v$ and $R = c_p - c_v$.

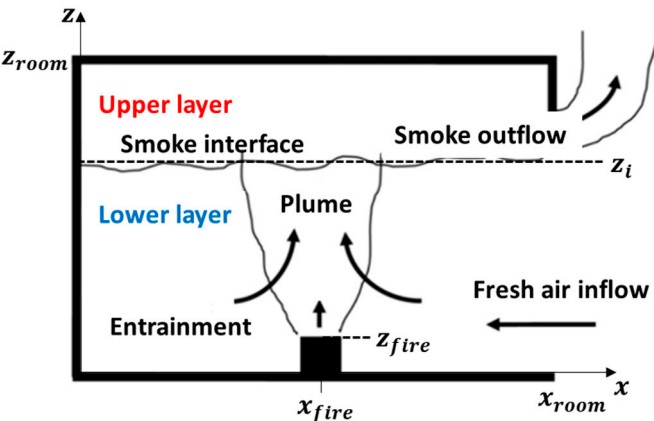

**Figure 1.** Schematic of the two-zone model in the plane $y = y_{fire}$.

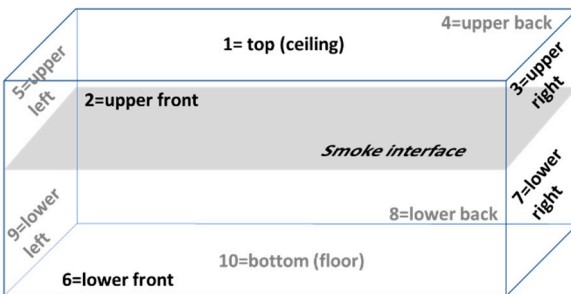

**Figure 2.** Schematic used for the ten-surface model [12] and notations.

The two-zone model is formulated as a set of ordinary differential equations given below, along with the closure relationships.

### 2.2. Governing Equations

Following the two-zone modeling concept, the gas in each layer k has attributes of mass, density, temperature, volume, and pressure denoted, respectively, $m_k$, $\rho_k$, $T_k$, $V_k$,

and $p_k$, where $k = u$ for the upper layer and $k = l$ for the lower layer. Some relationships exist between these variables. For example:

$$V_u + V_l = V_{room} \ (total\ volume) \tag{1}$$

$$\rho_k = m_k / V_k \ (density) \tag{2}$$

$$E_k = c_v m_k T_k \ (internal\ energy) \tag{3}$$

$$p_k V_k = m_k R T_k \ (ideal\ gas\ law) \tag{4}$$

The mass conservation equations for each species s of the gas mixture, which is composed of $N_s$ species, can be written in the upper and lower zones as:

$$\frac{dm_{su}}{dt} = \dot{m}_s^{ent} + \dot{m}_s^{pc} + \dot{m}_{s,u}^{ov} \qquad s = 1, N_s \tag{5}$$

$$\frac{dm_{sl}}{dt} = -\dot{m}_s^{ent} + \dot{m}_{s,l}^{ov} \qquad s = 1, N_s \tag{6}$$

where $\dot{m}_s^{ent}$ is the mass flow rate of species s in the air entrained by the fire plume, $\dot{m}_s^{pc}$ is the net rate of production/destruction of species s due to pyrolysis and combustion, and $\dot{m}_{s,k}^{ov}$ is the net mass flow rate of species s entering or leaving the layer k through vents due to natural (e.g., doors or windows) or mechanical ventilation.

The energy equations in the two layers are expressed as:

$$\frac{dE_u}{dt} = Q_c + c_p \dot{m}^b T_\infty + c_p \dot{m}^{ent} T_l + \dot{h}_u^{ov} - \sum_{w=1}^{5} \dot{q}_w^{conv} A_w + \dot{h}_u^{rad} - p_u \frac{dV_u}{dt} \tag{7}$$

$$\frac{dE_l}{dt} = -c_p \dot{m}^{ent} T_l + \dot{h}_l^{ov} - \sum_{w=6}^{10} \dot{q}_w^{conv} A_w + \dot{h}_l^{rad} - p_l \frac{dV_l}{dt} \tag{8}$$

where $Q_c$ is the convective fraction of the heat release rate (HRR) $Q$, $\dot{m}^b$ is the actual pyrolysis rate, $A_w$ is the area of the wall segment w, $\dot{h}_k^{ov}$ is the enthalpy source terms due to ventilation flows entering or leaving the layer k, $\dot{h}_k^{rad}$ is the net enthalpy due to radiation into the layer k, $\dot{q}_w^{conv}$ is the convective heat flux at the wall segment w, $T_\infty$ is the ambient temperature.

As previously mentioned, a momentum equation is added to calculate the displacement rate of the smoke layer interface $u_{sli}$. It is obtained by applying the variable-mass Newton's second law to the control volume extending from the interface to the ceiling (i.e., the upper layer). The Newton's law states that the sum of all forces that act upon the control volume is equal to the net rate of mechanical momentum relative to the control volume, which leads to:

$$\frac{dm_u u_{sli}}{dt} = \overline{u}^p(z_i)\dot{m}^p(z_i) - (p_l - p_u)A_{room} - g m_u \tag{9}$$

where $z_i$ is the interface height, $m_u = \sum_{s=1}^{N_s} m_{s,u}$, $A_{room} = x_{room} \, y_{room}$, $\overline{u}^p(z_i)$ is the average plume velocity at the interface height, and $\dot{m}^p = \dot{m}^b + \dot{m}^{ent}$ is the plume mass flow rate. The terms on the left-hand side of Equation (9) correspond, respectively, to the forces exerted on the interface by the plume, the pressure gradient, and gravity, respectively.

The volumes of the two layers are calculated from the following relations:

$$\dot{V}_l = -\dot{V}_u = u_{sli} A_{room} \tag{10}$$

The layer interface height is thus given by: $z_i = V_l / A_{room}$.

Due to the CFL limitation of the time step (typically about 50 µs), the ordinary differential Equations (5)–(10) are solved by using the simple explicit Euler method. An iterative

procedure is also used, strengthening the coupling between equations, and ensuring that the mass and energy conservation are satisfied at each time step.

*2.3. Source Terms*

2.3.1. Plume Entrainment

Following CFAST [5], Heskestad's correlation [13] is used to evaluate the mass entrained by the fire/plume from the lower layer to the upper layer at a height z above the base of the fire. The equations of Heskestad's plume are as follows:

- Above the mean flame height $z_L$:

$$\dot{m}^{ent}(z) = C_1 Q_C^{1/3}(z-z_0)^{5/3}\left[1 + C_2 Q_C^{2/3}(z-z_0)^{-5/3}\right] \tag{11}$$

- At and below the flame height, mass flow rates in fire plumes have been found to increase linearly with height [14], from zero at the fire base to the flame-tip value, leading to:

$$\dot{m}^{ent}(z) = \dot{m}^{ent}(z_L)z/z_L \tag{12}$$

where $Q_c$ is in kW and $c_p$ in kJ/kg/K, $z_L = D(-1.02 + 3.7Q^{*0.4})$, $z_0 = D(-1.02 + 1.4Q^{*0.4})$, and $Q^* = Q/(c_p\rho_\infty T_\infty\sqrt{gD}D^2)$. $D$ is the diameter of the fire source (or effective diameter for noncircular fire sources such that $\pi D^2/4$ is the area of the base of the fire). The constants $C_1$ and $C_2$ are given by: $C_1 = 0.196(g\rho_\infty^2/c_p T_\infty)^{1/3}$ and $C_2 = 2.9/(\sqrt{g}c_p\rho_\infty T_\infty)^{2/3}$. The subscript $\infty$ refers to ambient conditions. For most fire calculations, it is accurate enough to neglect the effect of the change of the molecular weight from that of air, so that density is determined primarily by its temperature: $\rho_\infty T_\infty = 353$ kg·K/m$^3$ [15]. For weak plumes, a limit on the mass entrainment is introduced [5]: $\dot{m}^{ent}(z) < Q_C/[c_p(T_u - T_l)]$.

In [13], Heskestad assumed that the plume velocity profile at a given height z, $u^p(r,z)$ can be represented as a Gaussian function of the plume radius r. Therefore, the mean plume velocity $\overline{u}^p(z_i)$ at the interface height that appears in the momentum equation can be obtained by integrating the velocity Gaussian function between $r = 0$ and the plume radius where the gas velocity has declined to half the value at the centerline. This gives: $\overline{u}^p(z_i) = 0.5\pi u_0^p(z_i)\sigma_u^2(z_i)$, where $\sigma_u(z_i)$ is a measure of the plume width at $z = z_i$, as defined in [5].

2.3.2. Vent Flow

Natural Flow through Vertical Vents (e.g., Doors or Windows)

Natural flow at a vent is governed by the pressure difference $\Delta p$ between the two sides of the vent. In the present model, the vent flow is calculated by integrating Bernoulli's equation along the vertical direction with the correct number of neutral planes (i.e., the elevation at which $\Delta p = 0$ or at which flow reversal occurs). Figure 3 depicts an example of vent flow through an opening with a neutral plane below the interface.

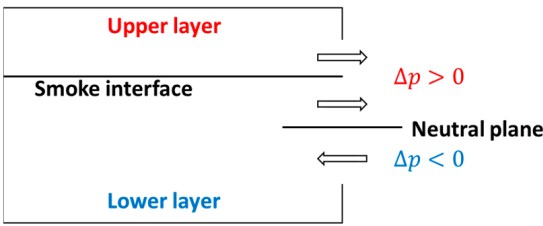

**Figure 3.** Example of vertical-vent configuration.

The approach to calculating the vent flow consists of partitioning the opening in one or more vertical slabs where each slab is bounded by the interface, the neutral plane, and the vent boundaries [15]. In the example of Figure 3, there are three slabs, one between

the lower vent boundary and the neutral plane, one between the neutral plane and the interface, and one between the interface and the higher vent boundary.

The mass flow for each slab is determined from [15]:

$$\dot{m}_{slab} = \frac{\sqrt{8\rho^*}}{3} C_{vv} A_{slab} \frac{x^2 + xy + y^2}{x + y} \tag{13}$$

where $x = \sqrt{|\Delta p_t|}$, $y = \sqrt{|\Delta p_b|}$, $C_{vv}$ is the constriction (or flow) coefficient (here, $C_{vv} = 0.68$), and $A_{slab}$ is the cross-sectional area of the slab. $\Delta p_t$ and $\Delta p_b$ are the pressure differences at the top and bottom elevations of the slab, respectively. The pressure difference at a height z is given by:

$$\Delta p(z) = \begin{cases} p_l + 0.5\rho_l g z_i - \rho_l g z - p_{ext}(z) \ if \ z \leq z_i \\ p_l - 0.5\rho_l g z_i - \rho_u g(z - z_i) - p_{ext}(z) \ if \ z > z_i \end{cases} \tag{14}$$

where $p_{ext}(z) = p_0 - \rho_\infty g z$. The pressure $p_0$ represents the base (reference) pressure at the floor ($z = 0$), outside the compartment.

The density $\rho^*$ is calculated as follows, depending on the slab location $z_{slab} = (z_t + z_b)/2$ and the sign of $\Delta p_{slab} = (\Delta p_t + \Delta p_b)/2$:

$$\rho^* = \begin{cases} \rho_u \ if \ z_{slab} > z_i \ and \ \Delta p_{slab} > 0 \\ \rho_l \ if \ z_{slab} \leq z_i \ and \ \Delta p_{slab} > 0 \\ \rho_\infty \ if \ \Delta p_{slab} \leq 0 \end{cases} \tag{15}$$

Natural Flow through Horizontal Vents (e.g., Ceiling Hatches or Holes)

For a ceiling vent, the standard vent-flow model is used:

$$\dot{m}_{vv} = C_{vv} A_{vv} \begin{cases} \rho_u \sqrt{2\Delta p / \rho_u} \ if \ \Delta p > 0 \\ \rho_\infty \sqrt{2|\Delta p| / \rho_\infty} \ if \ \Delta p \leq 0 \end{cases} \tag{16}$$

where $\Delta p = p_u - p_{ext}(z_{room})$, $C_{vv}$ is the flow coefficient, and $A_{vv}$ is the vent area.

A similar expression is used for a floor vent. Note that the standard model always yields a unidirectional flow through the vent. Further development is required for horizontal vents in which the upward and downward mass flow rates depend on both pressure and density differences. Cooper's correlation [16] could be used to model possible bidirectional flow through a horizontal vent.

Forced Flow

Mechanical ventilation is considered. Through the vent, smoke can be extracted out of the compartment to ambient, or ambient air can be supplied into the compartment. The mass flow rate due to forced convection through a vent located in the layer k, neglecting the effects of natural convection, can be deduced from the general equation for subsonic flow of an ideal gas [15]:

$$\dot{m}_{fv} = C_{fv} A_{fv} \begin{cases} \rho_k \left(\frac{p_{fv}}{p_k}\right)^{1/\gamma} \left\{ 2RT_k \frac{\gamma}{\gamma-1} \left[1 - \left(\frac{p_{fv}}{p_k}\right)^{(\gamma-1)/\gamma}\right] \right\}^{1/2} \ if \ \Delta p > 0 \\ \rho_{fv} \left(\frac{p_k}{p_{fv}}\right)^{1/\gamma} \left\{ 2RT_{fv} \frac{\gamma}{\gamma-1} \left[1 - \left(\frac{p_k}{p_{fv}}\right)^{(\gamma-1)/\gamma}\right] \right\}^{1/2} \ if \ \Delta p \leq 0 \end{cases} \tag{17}$$

where $\Delta p = p_k - p_{fv}$, $C_{fv}$ is the vent discharge coefficient, and $A_{fv}$ is the vent area. When air is supplied into the compartment, the user specifies $T_{fv}$, and $\rho_{fv} = p_{fv}/RT_{fv}$. The initial volumetric flow rate is also specified by the user, which allows us to calculate the terms $C_{fv}$ and $p_{fv}$, prior to fire simulations.

### 2.3.3. Conduction

The one-dimensional heat conduction equation is solved to calculate conduction heat transfer within wall segments:

$$\rho_w c_w \frac{\partial T_w}{\partial t} = \frac{\partial}{\partial x}\left(\lambda_w \frac{\partial T_w}{\partial x}\right) \tag{18}$$

with the associated initial and boundary conditions:

$$t = 0, \ 0 \leq x \leq e, \ T_w(x,0) = T_\infty \tag{19}$$

$$Inner\ surface, \ x = 0, \ -\lambda_w \frac{\partial T_w}{\partial x}\bigg|_{x=0} = \dot{q}_w^{conv}\bigg|_{x=0} + \dot{q}_w^{rad}\bigg|_{x=0} \tag{20}$$

$$Outer\ surface, \ x = e, \ \lambda_w \frac{\partial T_w}{\partial x}\bigg|_{x=e} = \dot{q}_w^{conv}\bigg|_{x=e} + \dot{q}_w^{rad}\bigg|_{x=e} \tag{21}$$

where $e$ is the wall segment thickness, $\lambda_s$, $\rho_s$, and $c_s$ are the thermal conductivity, density, and heat capacity of the wall segment, $x$ refers to the direction normal to the wall segment, and $\dot{q}_w^{conv}$ and $\dot{q}_w^{rad}$ are the convective and net radiative heat fluxes at the wall segment surface.

The heat conduction equation is solved in the direction normal to the wall surfaces using a uniform mesh of cell-centered control volumes. The spatial derivatives are approximated by a second-order central difference scheme, the time derivative by a semi-implicit time marching scheme. If the wall element is composed of multiple layers of different materials, the resolution is unchanged, but the thermal conductivity at the interface between these materials is calculated using a harmonic mean [17]. The tri-diagonal system of algebraic equations from discretization is solved using the well-known Thomas algorithm [17]. Due to the nonlinear coupling between temperature and radiative heat transfer, an iterative technique is used to solve the conjugate problem.

### 2.3.4. Radiation

The model calculates the heat transfer by radiation between the fire, the gas layers, and the 10 wall segments (source terms in the energy equations, $\dot{h}_u^{rad}$ and $\dot{h}_l^{rad}$, and in the conduction equation, $\dot{q}_w^{rad}\big|_{x=0}$) using the method developed in [12]. It also considers the radiative contribution (emission and absorption) of soot and gaseous species $CO_2$ and $H_2O$, as done in CFAST [5]. The net radiative heat flux from the outside surface of the wall segment $w$ is: $\dot{q}_w^{rad}\big|_{x=e} = \varepsilon_w \sigma \left(T_\infty^4 - T_w\big|_{x=e}^4\right)$, where $\sigma = 5.67 \times 10^{-8} \ \mathrm{W/m^2/K^4}$ is the Stefan-Boltzman constant and $\varepsilon_w$ is the surface emissivity.

### 2.3.5. Convection

The transfer of heat between the gas and walls is handled differently at the ceiling, upper and lower wall segments, and floor. It depends on the position of the fire in the compartment, the orientation of the wall segment and the presence of the ceiling jet.

Standard Convection (No Ceiling Jet Effect)

The convective heat flux at the inner surface of the wall segment w is given by:

$$\dot{q}_w^{st} = h(T - T_w) = h\Delta T_w \tag{22}$$

where $T$ is the gas temperature and $T_w$ is the temperature of the inner or outer surface of the wall segment. The convection coefficient is defined as $h = kNu_L/L$, where $k$ is the thermal conductivity of the gas and $L$ is the characteristic length of the geometry. The Nusselt number $Nu_L$ is based on the Rayleigh number $Ra_L = g\beta|\Delta T_w|L^3/\nu\alpha$, where $\beta = 1/\overline{T}_\beta$ is the volumetric expansion coefficient evaluated at the temperature $\overline{T}_\beta = (T + T_w)/2$, $\nu$ is

the kinematic viscosity, and $\alpha$ is the thermal diffusivity. The typical correlations applicable to the problem at hand are available in the literature [18–20]. Table 1 gives the correlations used in the model in the absence of ceiling jet.

**Table 1.** Empirical correlations used for convection heat transfer [18–20].

| Geometry | Correlation | Comments |
|---|---|---|
| Side walls | $Nu_L^{1/2} = 0.825 + \dfrac{0.387 Ra_L^{1/6}}{\left[1 + (0.492/Pr)^{9/16}\right]^{8/27}}$ | Lower wall segments : $L = z_i$ Upper wall segments: $L = z_{room} - z_i$ $\overline{T}_f = T_w - 0.25(T_w - T)$ |
| Ceiling for $\Delta T_w > 0$ and floor for $\Delta T_w \leq 0$ | $Nu_L = 0.54 Ra_L^{1/4}$ for $10^4 \leq Ra_L \leq 10^7$ $Nu_L = 0.15 Ra_L^{1/3}$ for $10^7 \leq Ra_L \leq 10^{11}$ | $L = A_w/P$ where $P$ is the perimeter of the wall segment and |
| Ceiling for $\Delta T_w \leq 0$ and floor for $\Delta T_w > 0$ | $Nu_L = 0.27 Ra_L^{1/4}$ | $\overline{T}_f = (T + T_w)/2$ |

The Prandtl number is defined as $Pr = \nu/\alpha$. Except $\beta$, gas properties are assumed to be those of air at the film temperature $\overline{T}_f$:

$$\nu = 0.04128 \times 10^{-7} \overline{T}_f^{5/2} / \left(\overline{T}_f + 110.4\right) \tag{23}$$

$$k = 2.72 \times 10^{-4} \overline{T}_f^{4/5} \tag{24}$$

$$\alpha = 10^{-9} \overline{T}_f^{7/4} \tag{25}$$

Convective Heating Due to Ceiling Jet Effect

In case of a fire, the flame and plume spread vertically upward and can impinge the ceiling, forming a ceiling jet that extends radially and, when it is blocked by the walls, forms a downward-spinning wall jet flow that is eventually turned back inward and upward by its own buoyancy [16]. The heat transfer at the ceiling is then driven by the temperature and velocity of the ceiling jet. The Cooper correlation [16] is used to evaluate the local convective heat flux between the plume and the inner ceiling surface:

$$\dot{q}_{ceil} = h(T_{ad} - T_{ceil}) \tag{26}$$

where $T_{ceil}$ is the temperature of the inner ceiling surface and $T_{ad}$ is a characteristic temperature that would be measured adjacent to an adiabatic inner ceiling surface.

In Equation (26), $h$ and $T_{ad}$ satisfy:

$$\frac{h}{\tilde{h}} = \begin{cases} 8.82 Re_H^{-1/2} Pr^{-2/3} \left[1 - (5 - 0.284 Re_H^{0.2}) r^*\right]; & 0 \leq r^* < 0.2 \\ 0.283 Re_H^{-0.3} Pr^{-2/3} r^{*-1.2} (r^* - 0.0771)/(r^* + 0.279); & 0.2 \leq r^* \end{cases} \tag{27}$$

$$(T_{ad} - T_u) / \left(T_u Q_H^{*2/3}\right) = \begin{cases} 10.22 - 14.9 r^*; & 0 \leq r^* < 0.2 \\ 8.39 f(r^*); & 0.2 \leq r^* \end{cases} \tag{28}$$

where $r^* = r/H$ is a geometric parameter defined as the ratio between the radial distance from fire plume axis, $r = \left[\left(x - x_{fire}\right) + \left(y - y_{fire}\right)\right]^{1/2}$, and the vertical distance between the ceiling and the (presumed) point source fire, $H = z_{room} - z'_{source}$.

The correlation involves the following terms:

$$Re_H = (gH)^{1/2} H Q_H^{*\,1/3} / \nu_u \tag{29}$$

$$Q_H^* = Q' / \left[\rho_u c_p T_u (gH)^{1/2} H^2\right] \tag{30}$$

$$\widetilde{h} = \rho_u c_p (gH)^{1/2} Q_H^{*1/3} \tag{31}$$

$$f(r^*) = \left(1 - 1.1r^{*0.8} + 0.808r^{*1.6}\right) / \left(1 - 1.1r^{*0.8} + 2.2r^{*1.6} + 0.69r^{*2.4}\right) \tag{32}$$

with

$$z'_{source} = \begin{cases} z_i - (z_i - z_{eq})\alpha^{3/5}\dot{M}^{*2/5}[(1+\sigma)/\sigma]^{1/5}; & z_i > z_{fire} \\ z_{fire}; & z_i \leq z_{fire} \end{cases} \tag{33}$$

$$Q' = \begin{cases} Q_c \sigma \dot{M}^* / (1+\sigma); & z_i > z_{fire} \\ Q_c; & z_i \leq z_{fire} \end{cases} \tag{34}$$

$$\sigma = \left(1 - \alpha + 9.115 Q_{eq}^{*2/3}\right) / (\alpha - 1) \tag{35}$$

$$\alpha = T_u / T_l \tag{36}$$

$$Q_{eq}^* = \left[0.21 Q_c / \left(c_p T_l \dot{m}^p\right)\right]^{3/2} \tag{37}$$

$$z_{eq} = z_i - \left[Q_c / \left(Q_{eq}^* \rho_l c_p T_l g^{1/2}\right)\right]^{2/5} \tag{38}$$

$$\dot{M}^* = \begin{cases} \left(1.04599\,\sigma + 0.360391\,\sigma^2\right) / \left(1 + 1.37748\,\sigma + 0.360391\,\sigma^2\right) & if\ \sigma > 0 \\ 0 & if\ -1 < \sigma \leq 0 \end{cases} \tag{39}$$

In these relations, $z_{fire}$ is the elevation of the base of fire (Figure 1), and the kinematic viscosity $\nu_u$ and Prandtl number are evaluated at temperature $T_u$.

The average convective flux is obtained by integrating the above relationship over the ceiling area:

$$\dot{q}_{w=1}^{cj} = \frac{1}{x_{room} y_{room}} \int_0^{x_{room}} \int_0^{y_{room}} \dot{q}_{ceil}(x,y) dx dy \tag{40}$$

As mentioned previously, the ceiling jet can be blocked by the relatively cool wall surfaces, which can increase the rate of heat transfer to the side wall surfaces. The model of Cooper [16] is then used to calculate the ceiling-jet-induced convective heat flux for the upper and lower wall segments, $\dot{q}_{w=2to9}^{cj}$.

The following strategy is adopted for estimating the average rate of convective heat transfer to the wall segments of the compartment:

- For all wall segments, calculate $\dot{q}_w^{st}$ from Equation (22).
- For all wall segments, except the floor ($1 \leq w \leq 9$):

  ○ In case of convective heating ($\Delta T_w > 0$), calculate $\dot{q}_w^{cj}$ from the Cooper's model and use the modified convective heat flux $\dot{q}_w^{conv} = max\left(\dot{q}_w^{st}, \dot{q}_w^{cj}\right)$;

  ○ In case of convective cooling ($\Delta T_w \leq 0$), use $\dot{q}_w^{conv} = \dot{q}_w^{st}$.

### 2.3.6. Combustion
#### Single-Fuel Combustion

As in CFAST [5], the combustion of a single fuel with molecular formula $C_{n_C} H_{n_H} O_{n_O} N_{n_N} Cl_{n_{Cl}}$ is described by the one-step reaction:

$$C_{n_C} H_{n_H} O_{n_O} N_{n_N} Cl_{n_{Cl}} + \nu_{O_2} O_2 \rightarrow \nu_{CO_2} CO_2 + \nu_{H_2O} H_2O + \nu_{CO} CO + \nu_s Suies + \nu_{HCl} HCl + \nu_{HCN} HCN \tag{41}$$

The two-zone model tracks over time the eight species that appear in this equation, plus nitrogen ($N_s = 9$). Initially, the composition of each layer is fixed at ambient conditions. The oxygen and nitrogen mass fractions are set to 0.233 and 0.767, respectively. The mass fraction of water vapor in ambient air is specified by the user in terms of relative humidity and the mass fractions of oxygen and nitrogen are adjusted accordingly. All other gas species are initially zero. The user specifies the composition of the fuel molecule and the

yields of soot, CO and HCN, $y_s$, $y_{CO}$ and $y_{HCN}$, which are related to their stoichiometric coefficients as follows:

$$
\begin{aligned}
\nu_s &= & y_s M_f / M_C \\
\nu_{CO} &= & y_{CO} M_f / M_{CO} \\
\nu_{CO_2} &= & n_C - (\nu_{CO} + \nu_{HCN} + \nu_s) \\
\nu_{H_2O} &= & 0,5 \ [n_H - (\nu_{HCl} + \nu_{HCN})] \\
\nu_{O_2} &= & \nu_{CO_2} + 0,5 \ (\nu_{H_2O} + \nu_{CO} - n_O) \\
\nu_{HCl} &= & n_{Cl} \\
\nu_{HCN} &= & \min\left(n_N \ ; \ y_{HCN} M_f / M_{HCN}\right)
\end{aligned}
\tag{42}
$$

where $M_f$, $M_{CO}$, $M_C$ and $M_{HCN}$ are the molar masses of fuel, CO, soot (soot are assumed to be pure carbon), and HCN.

The heat released by the fire has a convective and a radiative component, respectively $Q_c = (1 - \chi_r) \, Q$ and $Q_r = \chi_r \, Q$, where $\chi_r$ is the fraction of the heat release rate emitted by radiation [21]. Using the prescribed pyrolysis rate of fuel $\dot{m}^{pyr}$, the chemical heat release rate is $Q = \dot{m}^{pyr} \Delta h_{ch}$, where $\Delta h_{ch}$ is the chemical (effective) heat of combustion. Tewarson provides measured values of $\Delta h_{ch}$ for a wide range of fuels [21]. It is recalled that the ratio of the chemical heat of combustion to net heat of complete combustion is defined as combustion efficiency.

When the fuel is burning, product species are produced in direct proportion to the pyrolysis rate (e.g., $\dot{m}^{pc}_{CO_2} = \dot{m}^{b} \nu_{CO_2} M_{CO_2} / M_f$), but this can be constrained by the available oxygen in the compartment. For an unconstrained fire, $\dot{m}^{b} = \dot{m}^{pyr}$, whereas for the constrained fire, $\dot{m}^{b} < \dot{m}^{pyr}$. The fuel-rich flammability limit is incorporated by limiting the HRR as the oxygen level decreases until a lower oxygen limit is reached:

$$
Q = \min\left(\dot{m}^{pyr} \Delta h_{ch}, \ \dot{m}^{ent} Y_{O_2} C_{LII} \Delta h_{O_2}\right)
\tag{43}
$$

where $Y_{O_2}$ is the oxygen mass fraction, $\Delta h_{O_2}$ the heat of combustion per unit mass of oxygen, taken to be 13.1 $MJ/kg$ [21], and $C_{LII}$ a smoothing function ranging from 0 to 1 [5]. The pyrolysis rate for a constrained fire becomes: $\dot{m}^{b} = Q/\Delta h_{ch}$, where $Q$ is deduced from Equation (43).

Multi-Fuel Combustion

One of the original features of the model is the ability to simulate the combustion of two or more fuels burning in the same room. To achieve this, the concept of surrogate fuel molecule (SFM) is introduced. The SFM is assumed to be made up of carbon, hydrogen, oxygen, nitrogen, and chlorine atoms whose numbers $n_C$, $n_H$, $n_O$, $n_N$, and $n_{Cl}$ vary over time according to the initial molecules and the combustion rates of the individual fuels.

For example, if two fuels, referenced 1 and 2, are likely to burn, the number of carbon atoms of the equivalent molecule will be calculated as follows:

$$
n^{eq}_C(t) = \frac{n^1_C \, \dot{m}^{pyr}_1(t) + n^2_C \, \dot{m}^{pyr}_2(t)}{\dot{m}^{pyr}(t)}
\tag{44}
$$

where $\dot{m}^{pyr}_1$ and $\dot{m}^{pyr}_2$ are the prescribed pyrolysis rates of fuels 1 and 2, $\dot{m}^{pyr}(t) = \dot{m}^{pyr}_1(t) + \dot{m}^{pyr}_2(t)$, and $n^1_C$ and $n^2_C$ are the numbers of carbon atoms of fuels 1 and 2. The same procedure is followed for the number of hydrogen, oxygen, nitrogen and chlorine atoms of the SFM.

The advantage of this formulation is that it respects the stoichiometric rules and does not change the combustion chemistry, even if limited by oxygen availability. However, the combustion properties of the SFM must be recalculated, namely the soot, CO and

HCN yields, the chemical heat of combustion and the fraction radiated by the flame. For a mixture of $N_f$ fuels, these are expressed as:

$$y_s^{eq}(t) = \frac{\sum_{f=1}^{N_f} y_{s,f}\, \dot{m}_f^{pyr}(t)}{\sum_{f=1}^{N_f} \dot{m}_f^{pyr}(t)} \quad for\ s = soot, CO,\ and\ HCN \tag{45}$$

$$\Delta h_{ch}^{eq}(t) = \frac{\sum_{f=1}^{N_f} \Delta h_{ch,f}\, \dot{m}_f^{pyr}(t)}{\sum_{f=1}^{N_f} \dot{m}_f^{pyr}(t)} \tag{46}$$

$$\chi_r^{eq}(t) = \frac{\sum_{f=1}^{N_f} \chi_{r,f}\, \Delta h_{ch,f}\, \dot{m}_f^{pyr}(t)}{\sum_{f=1}^{N_f} \dot{m}_f^{pyr}(t) \Delta h_{ch,f}} \tag{47}$$

Figure 4 shows the time evolution of the pyrolysis rate and SFM for a fire involving heptane ($C_7H_{16}$) and polymethyl-methacrylate or PMMA ($C_5H_8O_2$). At the initial moment, the SFM is of the form $C_{6.68}H_{14.74}O_{0.32}$. It then evolves according to the prescribed burning rates (Figure 4a). After approximately 510 s of fire, only the PMMA continues to burn, which explains that the SFM is then composed of 5 atoms of carbon, 8 atoms of hydrogen, and 2 atoms of oxygen (Figure 4b).

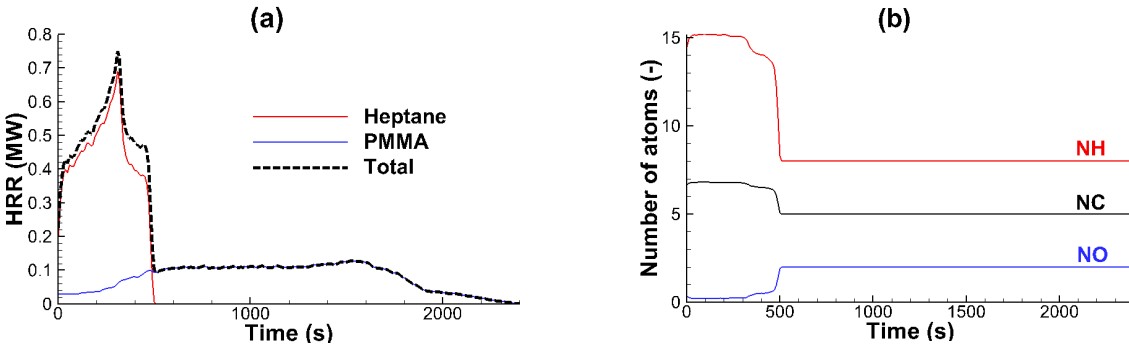

**Figure 4.** Time evolutions of (**a**) individual and total pyrolysis rates and (**b**) SFM, for a fire involving heptane and PMMA.

## 3. Experimental Setup and Fire Scenarios

To demonstrate the capability of the zone model in reproducing the consequences of a fire involving multiple fuels in a mechanically ventilated compartment, experiments have been conducted in the DIAMAN device of the IUSTI laboratory.

### 3.1. DIAMAN Device

As shown in Figures 5 and 6, DIAMAN consists of two cubic compartments with side length of 3 m. It has two airtight doors of 1 m × 2 m: one opening from compartment 2 to the outside, the other on the bulkhead separating the two compartments. The walls of the device, as well as the doors, are made of 1 cm thick steel, with the following thermal properties: a conductivity of 50 W/m/K, a density of 7800 kg/m³, a specific heat of 470 J/kg/K, and an emissivity of 0.7. The device has (a) four 200 mm × 300 mm rectangular Pyrex® viewing windows on the FRONT and BACK walls of each compartment, (b) four circular openings of 200 mm diameter for mechanical ventilation: two (low and high) intake vents on the WEST wall of compartment 1, one extraction vent at the ceiling of the two compartments, and (c) three openings of 800 mm × 300 mm, equipped with adjustable height guillotines, for natural ventilation: two on the WEST wall of compartment 1, and one on the bulkhead door. Figure 6 shows an exploded view of DIAMAN, showing the natural

ventilation openings and the exhaust ducts, in one of the two configurations studied (here, the bulkhead door is closed).

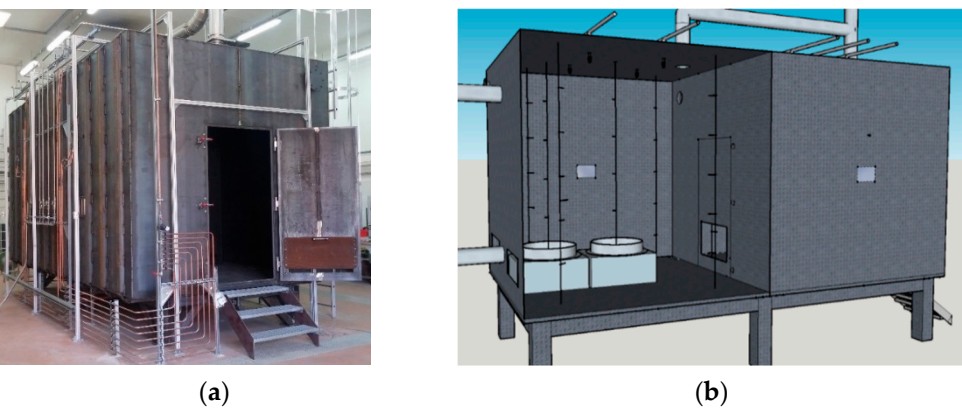

**Figure 5.** (**a**) Picture and (**b**) schematic of the DIAMAN device.

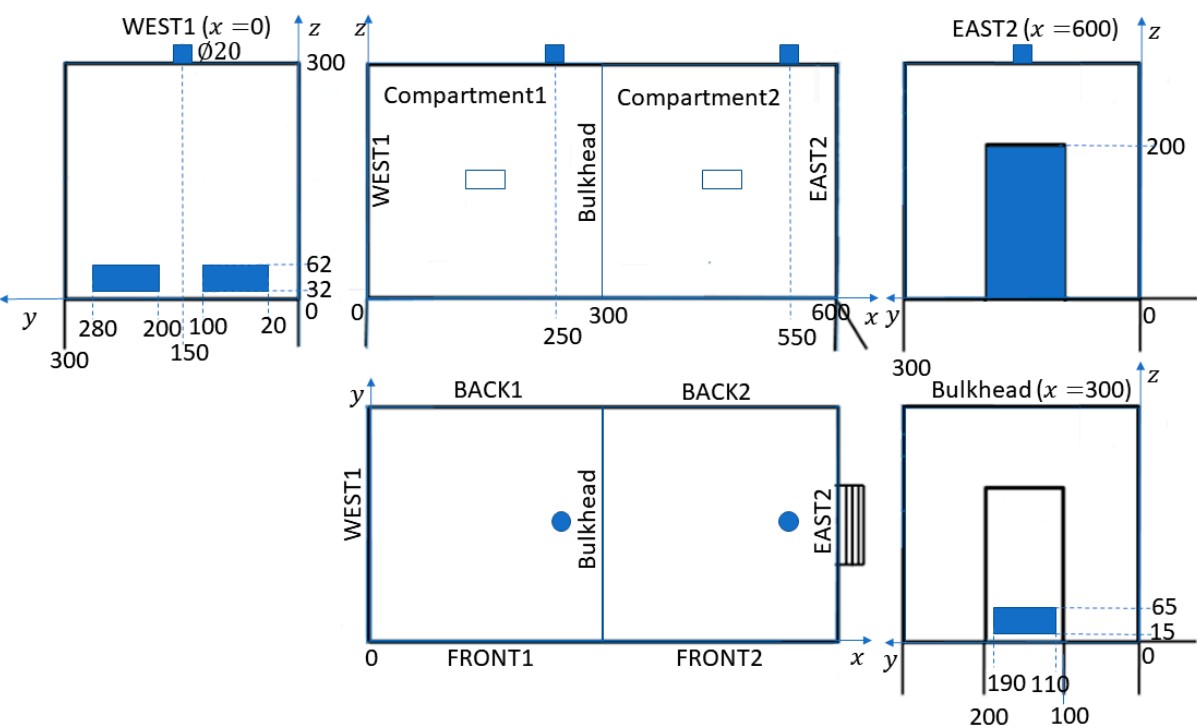

**Figure 6.** Exploded view of the DIAMAN device (dimensions in cm).

For all experiments, the fire was started in compartment 1, whose instrumentation includes:

- A CCD camera to observe the general behavior of the fire.
- Two SARTORIUS® electronic scales, placed in stainless steel thermally insulated boxes, for the measurement of fuel mass losses over time. They can support a maximum load of 150 kg, with an accuracy of 1 g and a response time of 0.1 s.
- Four trees of five K-type thermocouples of 1 mm diameter, positioned in the corners of the compartment, at 0.5 m from the vertical walls and at heights of 0.5, 1.0, 1.5, 2.0, and 2.5 m from the floor level (Figure 7a).
- Five surface K-type thermocouples, positioned on the outer surface of each wall of the compartment (Figure 7b).
- A pitot tube with integrated thermocouple in the exhaust duct to measure the volumetric flow rate and temperature of the exhausted gases.

- A TESTO-350 gas analyzer for measuring the concentrations of $O_2$ and $CO_2$ in the exhaust gases.
- A measurement of the static pressure in the compartment.

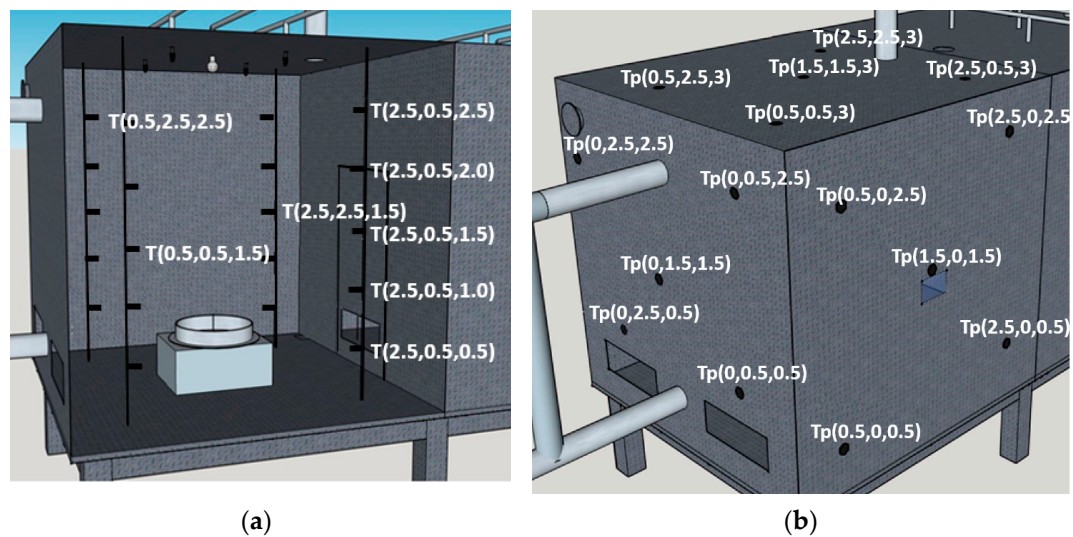

(**a**)                                        (**b**)

**Figure 7.** Schematics showing the locations of some thermocouples in compartment 1 to measure (**a**) gas and (**b**) wall temperatures.

### 3.2. Fuels

Four solid and liquid fuels, with different kinetics of pyrolysis and combustion, have been selected: heptane, the widely used thermoplastic PMMA (polymer poly-methyl methacrylate), dry untreated fir (DUF), and flexible polyurethane foam (PUF). They can be considered representative of the various fuels that can be found in most large structures such as office buildings, civilian or military ships, warehouses, etc. Fuel properties are given in Table 2.

**Table 2.** Fuel properties [21].

| Fuel | Chemical Formula | Density $(kg/m^3)$ | Yields of Fire Products (g/g) | | | $\Delta h_{ch}$ (MJ/kg) | $\chi_r$ |
|------|------------------|--------------------|------|------|------|--------------------------|----------|
| | | | Soot | CO | HCN | | |
| Heptane | $C_7H_{16}$ | 680 | 0.037 | 0.010 | 0 | 41.2 | 0.305 |
| PMMA | $C_5O_2H_8$ | 1160 | 0.022 | 0.010 | 0 | 24.2 | 0.302 |
| Dry untreated fir [1] | $C_6O_{10.2}H_{4.98}$ | 420 | 0.015 | 0.004 | 0 | 12.4 | 0.207 |
| PU foam [2] | $C_{6.3}O_{7.1}H_{2.1}N$ | 30 | 0.227 | 0.031 | 0 | 28.0 | 0.520 |

[1] assimilated to wood pine [21]. [2] The PU foam selected is a flexible, water-blown foam composed of two-thirds TDI (toluene diisocyanate) and one-third polyol. It does not contain a flame retardant. This type of foam is used for the filling of mattresses, seats of chairs, armchairs, or benches. It is here assimilated to GM23 flexible PU foam [21].

### 3.3. Fire Scenarios

The ten tests performed correspond to various combinations and loads of the fuels presented above, as well as two levels of containment of the device, depending on whether the bulkhead door was closed or open (Table 3). For all tests, the exhaust volumetric flow rate at the ceiling of compartment 1 was initially set to 800 $m^3$/h. There was no other mechanical ventilation. The openings on the WEST wall of compartment 1 were kept open, as well as the natural opening at the bottom of the bulkhead door when this door was closed.

**Table 3.** Fire scenarios.

| Test | Fuel(s) | Bulkhead Door |
|------|---------|---------------|
| 1 | DUF (8.46 kg)/PUF (2.3 kg) | |
| 2 | PMMA (7.37 kg)/PUF (2.33 kg) | |
| 3 | Heptane (5.34 kg)/PMMA (7.39 kg) | Closed |
| 4 | Heptane (5.37 kg)/PUF (2.33 kg) | |
| 5 | DUF (8.49 kg)/PMMA (7.41 kg) | |
| 6 | DUF (8.49 kg)/PUF (2.33 kg) | |
| 7 | PMMA (7.41 kg)/PUF (2.32 kg) | |
| 8 | Heptane (5.33 kg)/PMMA (7.39 kg) | Open |
| 9 | Heptane (5.37 kg)/PUF (2.29 kg) | |
| 10 | DUF (8.5 kg)/PMMA (7.4 kg) | |

The fire source was a combination of two of the following unit fuel fire sources (Figure 8):

- A 70 cm diameter pan filled with heptane;
- A wooden crib, formed by stacking 10 crisscrossed layers of 6 DUF sticks. Each stick was 0.5 m long, with a 3 cm square section. A holding grid was used to prevent the glowing sticks from falling out of the 70 cm diameter pan due to the collapse of the crib;
- Two PUF blocks, stacked in a 70 cm diameter pan. Each block has dimensions of 0.62 m × 0.62 m × 0.1 m;
- Four sheets of PMMA, arranged horizontally in a 70 cm diameter pan. Each sheet has dimensions of 0.2 m × 0.5 m × 0.03 m.

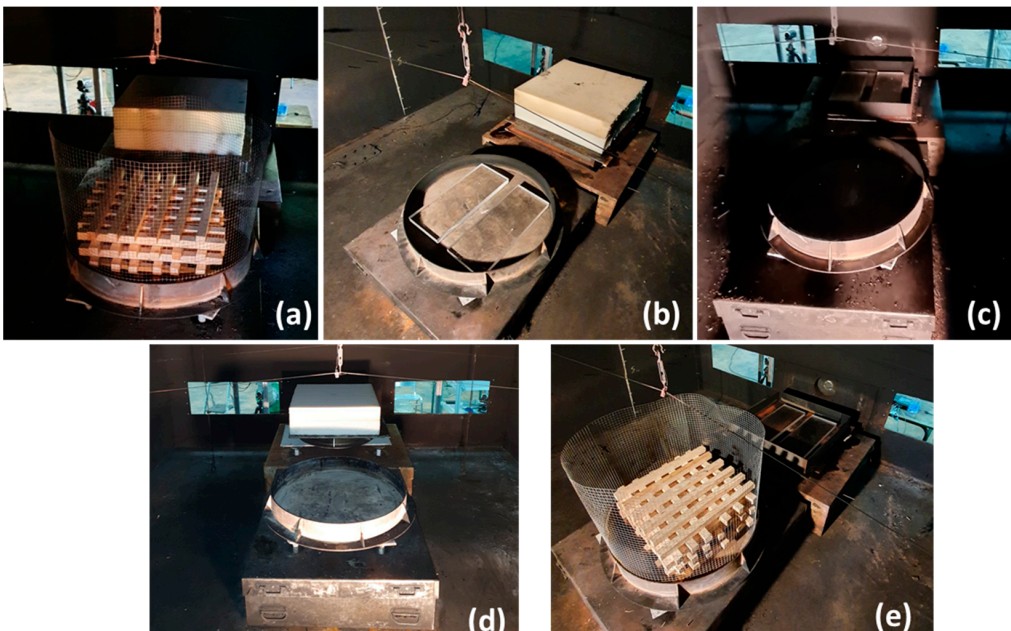

**Figure 8.** Fire sources involving (**a**) DUF and PUF (tests 1 and 6), (**b**) PMMA and PUF (tests 2 and 7), (**c**) heptane and PMMA (tests 3 and 8), (**d**) heptane and PUF (4 and 9), and (**e**) DUF and PMMA (tests 5 and 10).

For the tests using dry untreated fir and PMMA as fuel, the combustible elements were arranged in such a way that the flame generated by their combustion clings to the edge of the pan and occupies its entire surface, which makes it possible to assimilate the effective area of the fire source to that of the pan (0.385 m$^2$). Therefore, in any test, the total area of the multi-fuel fire source was 0.77 m$^2$.

As shown in Figure 5, the first fuel pan was placed in the center of compartment 1, while the second one was moved to the WEST wall, with its center 50 cm from the wall. Ignition of heptane and PUF was achieved by exposing the fuel to the flame of a gas burner for a few seconds. For DUF and PMMA, ignition was not done directly, but with a small amount of ethanol (0.4 kg) placed at the bottom of the pan.

## 4. Validation Results

### 4.1. Data Processing

The data of interest for the validation of the zonal model concern the evolution of the average gas and wall temperatures, and the volume concentrations of $CO_2$ and $O_2$ in the exhausted gases. The interface of the smoke layer is diffuse and cannot be measured with sufficient reliability, which is why the raw measurements are treated in a specific way.

The average temperature of the gas in the compartment is obtained by summing the local temperature values measured by the 20 thermocouples placed, as shown in Figure 7a, and dividing by 20. The value obtained is then compared to that calculated from the upper- and lower-layer temperatures and the elevation of the smoke interface predicted by the zone model: $\overline{T}^{mod} = [(z_{room} - z_i)T_u + z_i T_l]/z_{room}$.

Due to the distribution of the thermocouples on the walls of the compartment (Figure 7b), the average temperature of the side walls is calculated as follows:

$$\overline{T}_{sw}^{exp} = \frac{1}{4}\left[\overline{T}_{p_{FRONT}} + \overline{T}_{p_{BACK}} + \overline{T}_{p_{WEST}} + \overline{T}_{p_{EAST}}\right] \tag{48}$$

with

$$\overline{T}_{p_{FRONT}} = \frac{1}{6}\left[T_p(0.5,0,0.5) + T_p(2.5,0,0.5) + 2T_p(1.5,0,1.5) + T_p(0.5,0,2.5) + T_p(2.5,0,2.5)\right] \tag{49}$$

$$\overline{T}_{p_{BACK}} = \frac{1}{6}\left[T_p(0.5,3,0.5) + T_p(2.5,3,0.5) + 2T_p(1.5,3,1.5) + T_p(0.5,3,2.5) + T_p(2.5,3,2.5)\right] \tag{50}$$

$$\overline{T}_{p_{WEST}} = \frac{1}{6}\left[T_p(0,0.5,0.5) + T_p(0,2.5,0.5) + 2T_p(0,1.5,1.5) + T_p(0,0.5,2.5) + T_p(0,2.5,2.5)\right] \tag{51}$$

$$\overline{T}_{p_{EAST}} = \begin{cases} \frac{1}{6}\left[T_p(3,0.5,0.5) + T_p(3,2.5,0.5) + 2T_p(3,1.5,1.5) + T_p(3,0.5,2.5) + T_p(3,2.5,2.5)\right] \\ \qquad\qquad for\ Tests\ 1\ to\ 5 \\ \frac{1}{4}\left[T_p(3,0.5,0.5) + T_p(3,2.5,0.5) + T_p(3,0.5,2.5) + T_p(3,2.5,2.5)\right]\ for\ Tests\ 6\ to\ 10 \end{cases} \tag{52}$$

It is compared to that obtained from the predicted temperatures of wall segments 2 to 9: $\overline{T}_{sw}^{mod} = \sum_{w=2}^{9} T_w/8$.

For the ceiling, the average experimental temperature is calculated using a Gaussian profile about the center of the ceiling:

$$\overline{T}_{ceil}^{exp} = \frac{2\pi}{A_{w=1}} T_p(1.5,1.5,3) \int_0^R exp\left(-\frac{r^2}{\sigma^2}\right) r\,dr \tag{53}$$

where $R$ is the equivalent radius of the ceiling deduced from $R^2 = x_{room}y_{room}$, and $\sigma$ is estimated from thermocouple values as follows: $\sigma^2 = -2/ln\left(\overline{T}_{0.5}/T_p(1.5,1.5,3)\right)$, with $\overline{T}_{0.5} = 0.25\left[T_p(0.5,0.5,3) + T_p(0.5,2.5,3) + T_p(2.5,0.5,3) + T_p(2.5,2.5,3)\right]$. $\overline{T}_{ceil}^{exp}$ is then compared with the predicted ceiling temperature $T_{w=10}$.

The N-percentage rule by Cooper [22] is applied to estimate the smoke layer interface height from the measured vertical temperature profiles. Beforehand, the gas temperatures measured in the four corners at 1.0, 1.5, 2.0, and 2.5 were first averaged to obtain $\overline{T}_{1.0}$, $\overline{T}_{1.5}$, $\overline{T}_{2.0}$, and $\overline{T}_{2.5}$, as done previously for $\overline{T}_{0.5}$. Then, they were interpolated between

each thermocouple height to refine the temperature profile (e.g., every cm). The interface height $z_{int}$ is finally determined as the height value satisfying the following condition:

$$T(z_{int}, t) - T_0 = \frac{N}{100} \left[ max \left( \overline{T}_{2.5}(t) \right) - \overline{T}_{2.5}(0) \right] \tag{54}$$

The selection of the value of N is quite subjective. Values of 10, 15 and 20 have been suggested by Cooper [22]. Here, optimum values in the range of [15–40] are adopted depending on the test under study. Other techniques, namely the maximum gradient method by Emmons [15], the Janssens and Tran method [23], and the integral ratio method by He et al. [24], have been tested, but they were not able to describe the time evolution of the interface height with decreasing smoke layer thickness, especially at the end of burn.

### 4.2. Comparison Model/Experiments

Fire scenarios 1 to 10 have been simulated using the two-zone model. For all simulations, the time step was 50 µs. A run for the simulation of 1 h of fire takes less than 2 CPU minutes on a PC with Intel® Core i7-4770 (3.40 GHz) and 16 GB RAM.

Figures 9–18 show the individual and total HRR time histories and the comparison between numerical and experimental results for the ten tests, including the time evolution of the smoke layer interface height. The model captures well the experimental trends for all fire conditions, regardless of the confinement level of the enclosure.

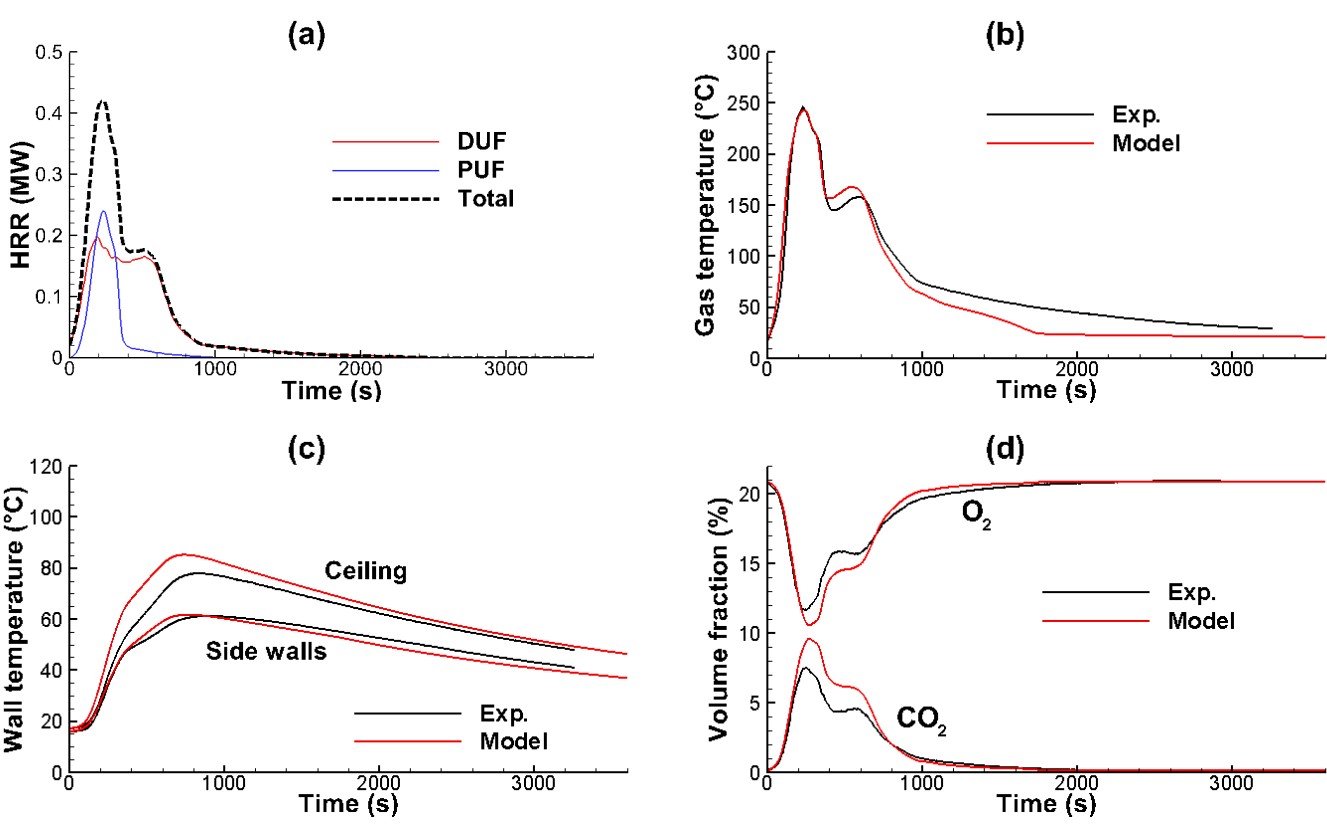

**Figure 9.** *Cont.*

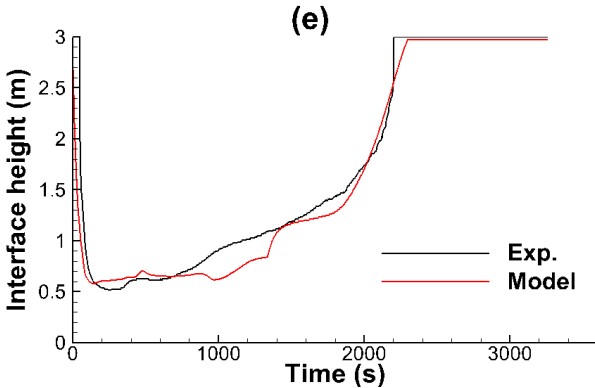

**Figure 9.** Test 1: (**a**) individual and total HRRs, (**b**) average gas temperature, (**c**) average wall temperatures, (**d**) volume fractions of $O_2$ and $CO_2$ in the exhausted gases, and (**e**) smoke layer interface height.

**Figure 10.** Test 2: (**a**) individual and total HRRs, (**b**) average gas temperature, (**c**) average wall temperatures, (**d**) volume fractions of $O_2$ and $CO_2$ in the exhausted gases, and (**e**) smoke layer interface height.

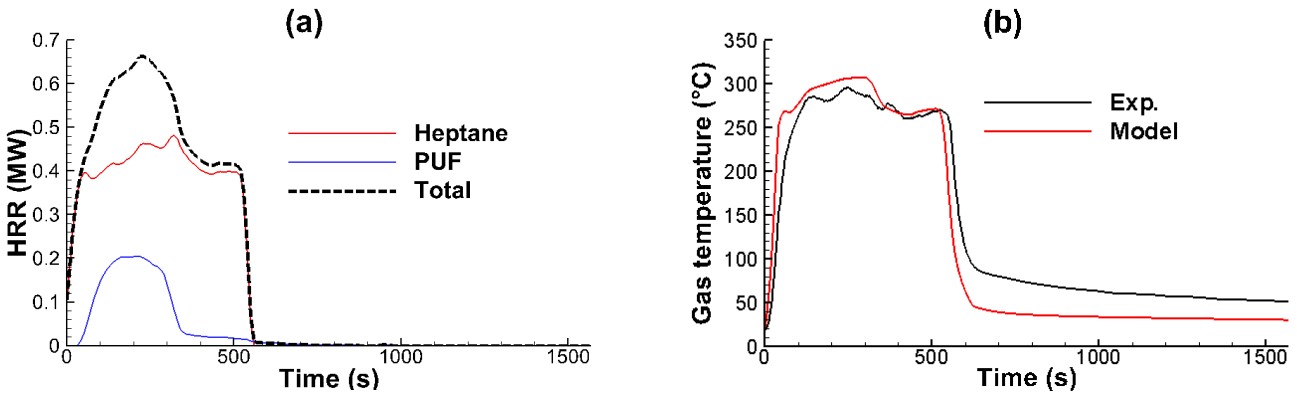

**Figure 11.** Test 3: (**a**) individual and total HRRs, (**b**) average gas temperature, (**c**) average wall temperatures, (**d**) volume fractions of $O_2$ and $CO_2$ in the exhausted gases, and (**e**) smoke layer interface height.

**Figure 12.** *Cont.*

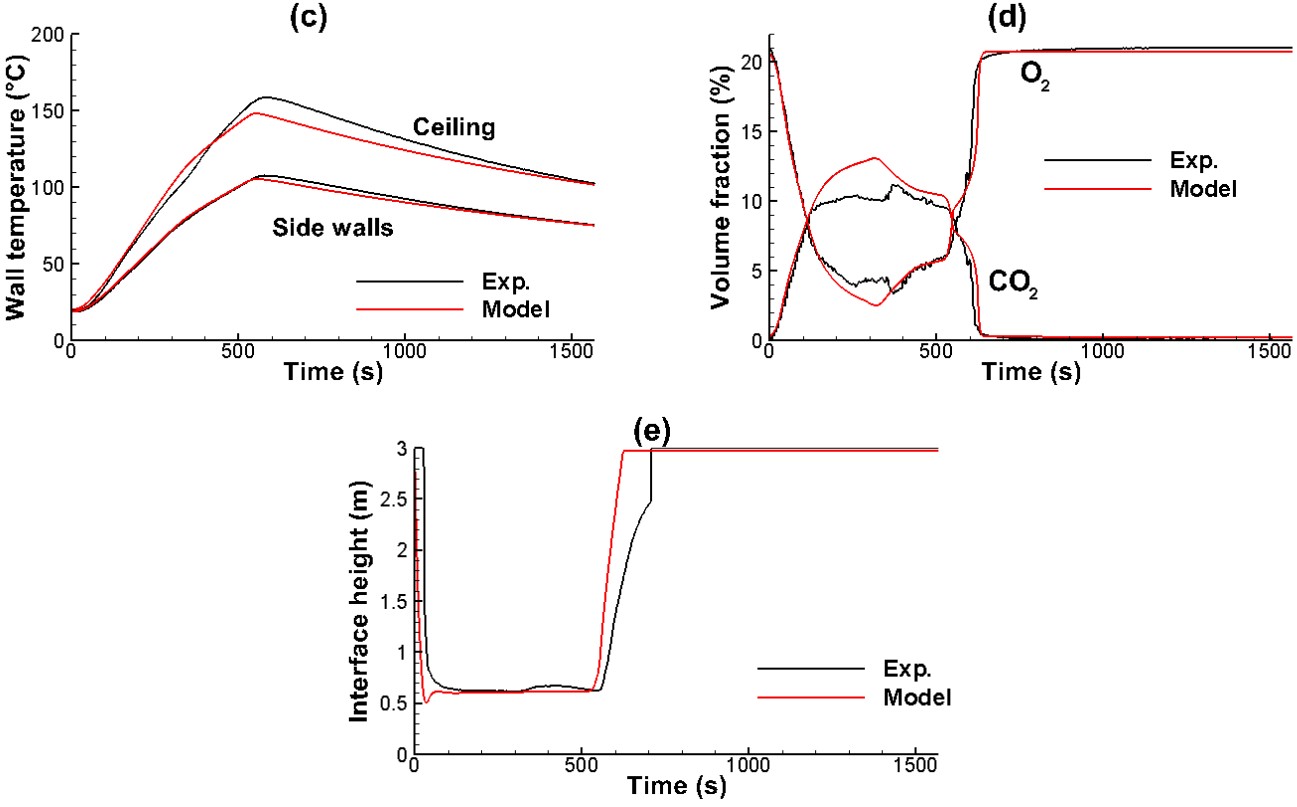

**Figure 12.** Test 4: (**a**) individual and total HRRs, (**b**) average gas temperature, (**c**) average wall temperatures, (**d**) volume fractions of $O_2$ and $CO_2$ in the exhausted gases, and (**e**) smoke layer interface height.

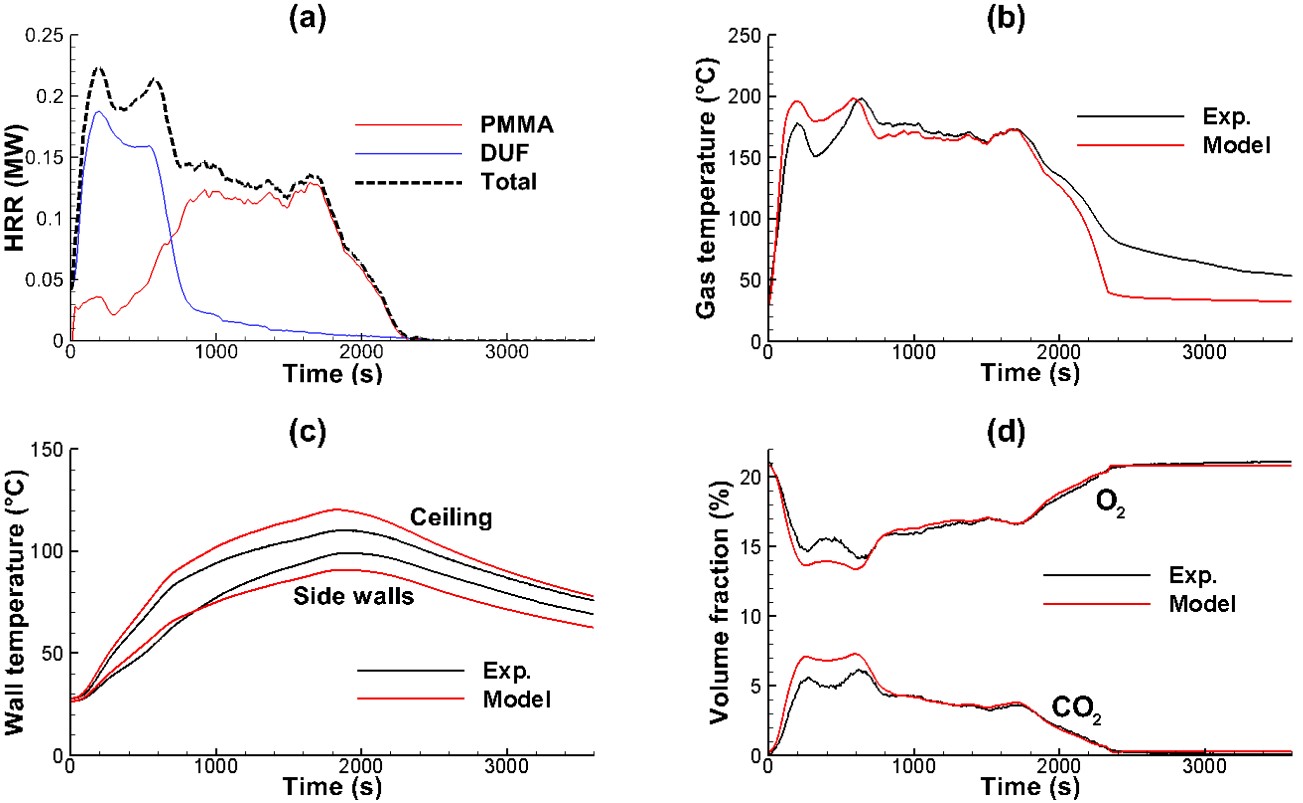

**Figure 13.** *Cont.*

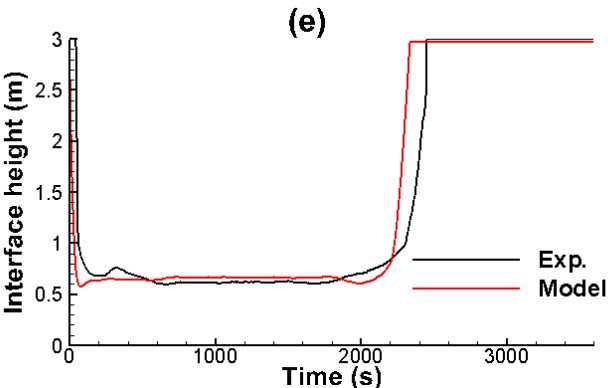

**Figure 13.** Test 5: (**a**) individual and total HRRs, (**b**) average gas temperature, (**c**) average wall temperatures, (**d**) volume fractions of $O_2$ and $CO_2$ in the exhausted gases, and (**e**) smoke layer interface height.

**Figure 14.** Test 6: (**a**) individual and total HRRs, (**b**) average gas temperature, (**c**) average wall temperatures, (**d**) volume fractions of $O_2$ and $CO_2$ in the exhausted gases, and (**e**) smoke layer interface height.

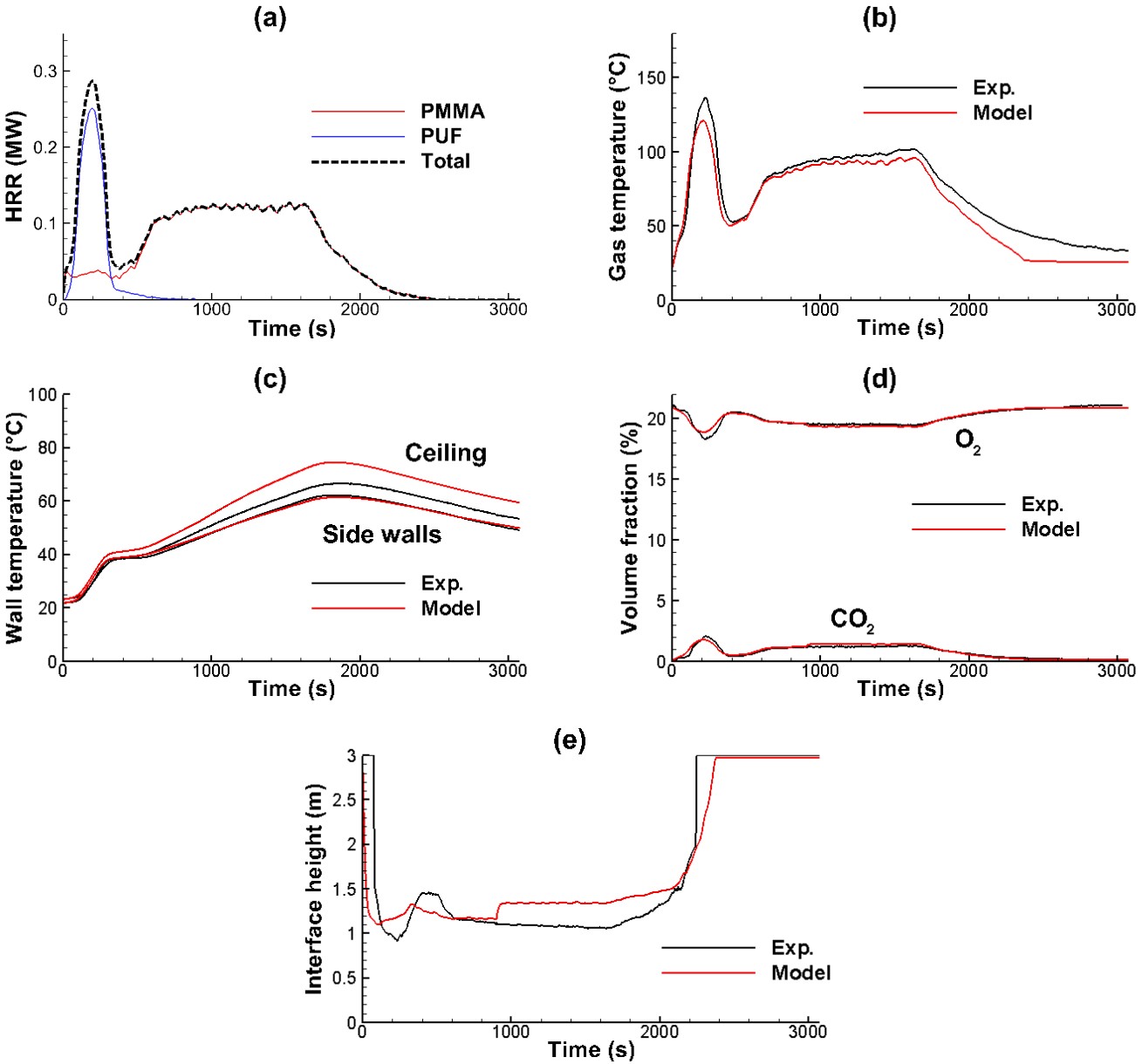

**Figure 15.** Test 7: (**a**) individual and total HRRs, (**b**) average gas temperature, (**c**) average wall temperatures, (**d**) volume fractions of $O_2$ and $CO_2$ in the exhausted gases, and (**e**) smoke layer interface height.

The predicted and experimental average gas temperatures versus time are similar in shape. The maximum level reached by the gas temperature and the time required to reach the peak are typically predicted within 12.5 and 20% of the experimental measurements, respectively. For test 9, the model underestimates the maximum value of the gas temperature. This can be attributed to not knowing the exact composition and combustion properties of the PUF we used (here, assimilated to GM23 foam). The maximum deviation in time to peak gas temperature is observed for test 4, with 20%, but this must be qualified due to the quasi-steady behavior of fire (Figure 12b). The overall comparison is more favorable to the model.

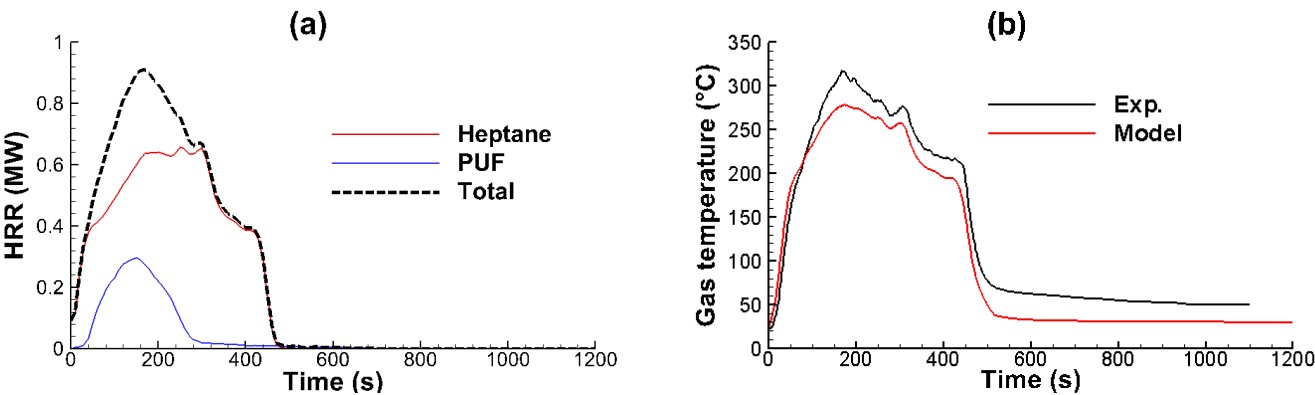

**Figure 16.** Test 8: (**a**) individual and total HRRs, (**b**) average gas temperature, (**c**) average wall temperatures, (**d**) volume fractions of $O_2$ and $CO_2$ in the exhausted gases, and (**e**) smoke layer interface height.

**Figure 17.** *Cont.*

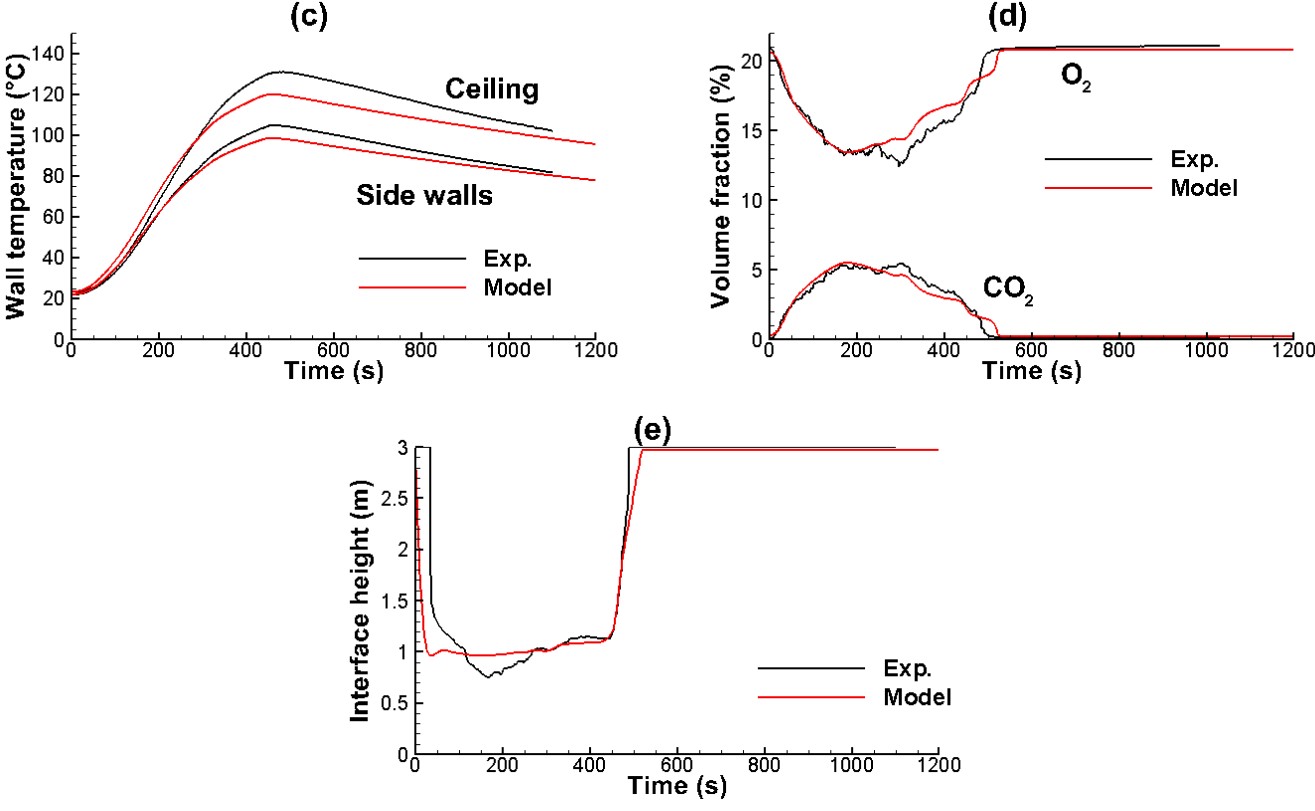

**Figure 17.** Test 9: (**a**) individual and total HRRs, (**b**) average gas temperature, (**c**) average wall temperatures, (**d**) volume fractions of $O_2$ and $CO_2$ in the exhausted gases, and (**e**) smoke layer interface height.

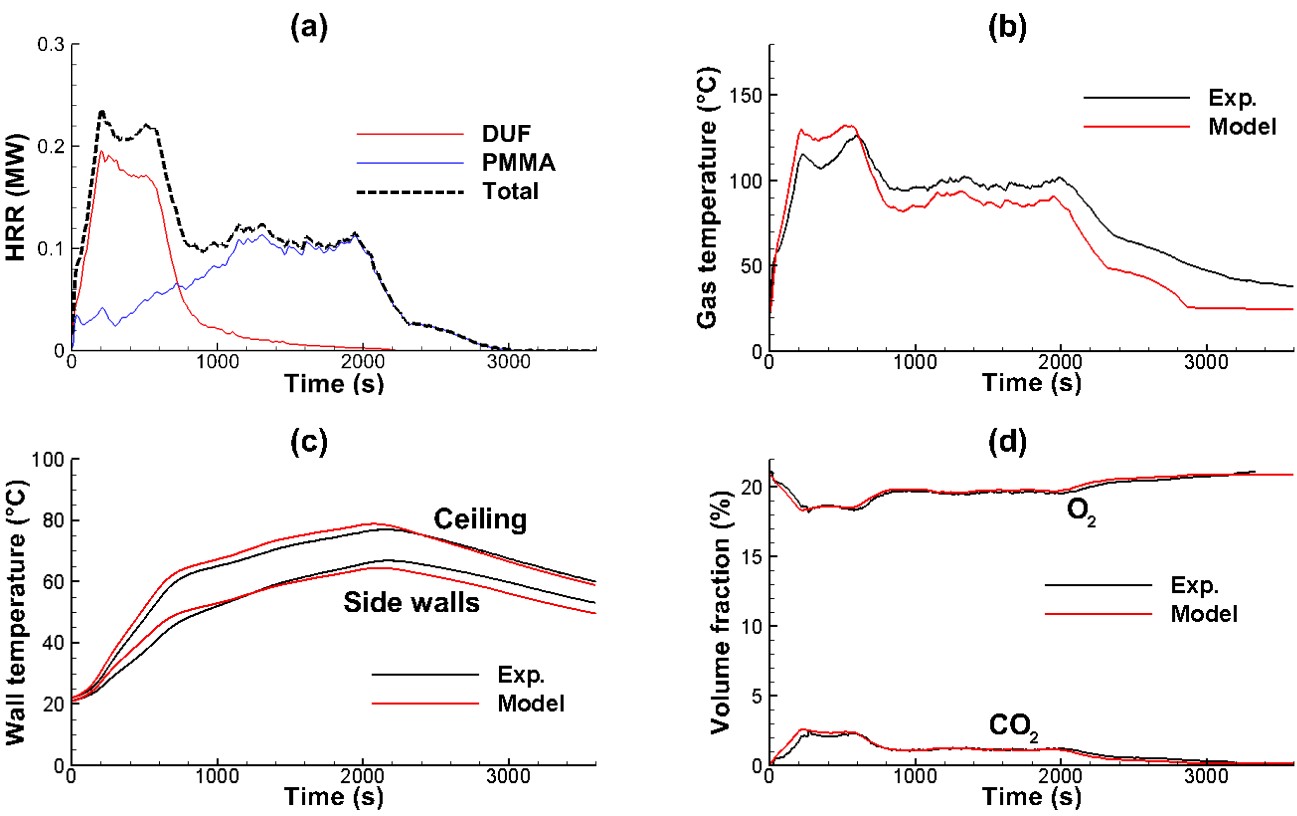

**Figure 18.** *Cont.*

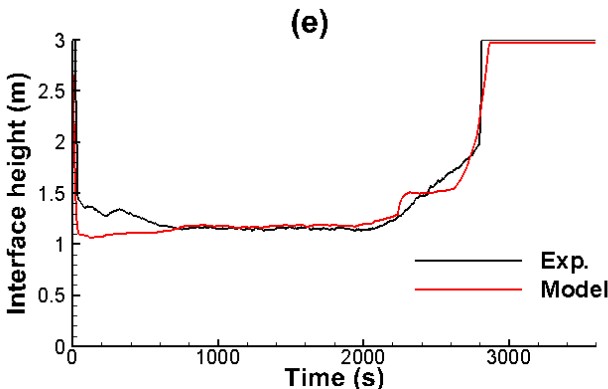

**Figure 18.** Test 10: (**a**) individual and total HRRs, (**b**) average gas temperature, (**c**) average wall temperatures, (**d**) volume fractions of $O_2$ and $CO_2$ in the exhausted gases, and (**e**) smoke layer interface height.

By comparing the predicted and measured wall temperatures, it is clear that the ceiling jet effect acts differently depending on the size of the fire. For relatively small fires, those that do not involve heptane, the model tends to overestimate the ceiling temperature, making it conservative. In contrast, when heptane is burning, the rate of convective heat release, and thus the ceiling temperature, is underestimated. This is particularly visible in the c-diagrams of Figures 11 and 16 for tests 3 and 8. The sidewall temperature is generally well reproduced by the model. The largest difference between the model and the experiment is observed for test 5, once the PMMA begins to burn. The strength of the ceiling jet is then not sufficient to allow the hot gases to flow back down the walls.

Combustion dynamics is well reproduced by the model (diagram d in Figures 9–18), which validates the concept of surrogate fuel molecule (SFM). For well ventilated fires (Tests 6 to 10), the model results, for both oxygen depletion and carbon dioxide production, are in good agreement with the experiments. When the bulkhead door is closed (Test 1 to 5), the agreement is less satisfactory, with overestimated oxygen depletion and carbon dioxide production. Much of this can be attributed to limited information on the combustion properties of the fuels involved in fire, especially DUF and PUF, and to the global one-step reaction mechanism with constant yields of soot and CO, when they are known to depend on the combustion conditions mainly in relation to the fuel-air equivalence ratio [25]. The HRR deficit induced by a higher oxygen consumption can explain a lower ceiling jet effect on the walls, as observed for tests 3 and 4. Note that the model reproduces well the return to the atmospheric oxygen level after burning is completed.

As shown in the e-diagrams of Figures 9–18, there is good agreement between the interface height predicted by the model and that deduced from temperature measurements. For tests 1 through 5, where the bulkhead door was kept closed, the smoke layer interface drops to about 60 cm in height, just above the fuel surface, and then rises again as the fire decays. For tests 6 to 10, the minimum elevation of the interface is about 1 m, as expected for such well-ventilated fires. Qualitative visual observations confirm these findings.

## 5. Conclusions

In this work, a two-zone model to simulate fire behavior and consequences in a compartment where multi-fuel combustion occurs was developed and validated using a series of full-scale compartment fire tests conducted in the IUSTI fire laboratory. In these experiments, different combinations of solid and liquid fuels and two levels of compartment confinement were studied, corresponding to both under-ventilated and well-ventilated fire conditions.

Conclusions were drawn by comparing simulation and experimental results:

- Regardless of the fuels used and the confinement level of the enclosure, the two-zone model reproduces the experimental trends well for all fire scenarios, including the time evolution of the smoke layer interface.
- The new concept of surrogate fuel molecule is a good alternative when several fuels are burning in the same compartment. For under-ventilated fires, differences between model results and measurements appear, which may be due to the limited information on the combustion properties of some fuels involved in fire and to the simple one-step reaction mechanism with constant yields of soot and CO.
- The impact of the ceiling jet on the walls depends strongly on the size of the fire. For small fires, the model is rather conservative. In contrast, it slightly underestimates the wall temperature increase for large fires whose flames and fire plumes can touch the ceiling.

Further modeling and numerical work should be conducted to improve the reaction mechanism and the ceiling jet modeling. For the latter, the flame height dependence could be studied. This study also provides data that other researchers could use in developing their fire model.

**Author Contributions:** B.P., conceptualization, methodology, and writing—original draft; N.S., software; Y.P., M.M., and P.P., investigation; J.L., project administration; N.D., validation; T.P., data curation. All authors have read and agreed to the published version of the manuscript.

**Funding:** This research was funded by the Association Nationale de la Recherche et de la Technologie (ANRT), CIFRE Grant n° 2017/1739, and jointly by the Délégation Générale de l'Armement (DGA) and the Agence Nationale de la Recherche (ANR), Grant n°ANR-17-ASMA-0005.

**Institutional Review Board Statement:** Not applicable.

**Informed Consent Statement:** Not applicable.

**Data Availability Statement:** Data supporting the results of this study are available on request from J.L.

**Acknowledgments:** The authors wish to express their thanks for the financial support of ANRT, DGA, and ANR.

**Conflicts of Interest:** The authors declare no conflict of interest. The funders had no role in the design of the study; in the collection, analyses, or interpretation of data; in the writing of the manuscript; or in the decision to publish the results.

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
