# Peer review of "Development and Validation of a Zone Fire Model Embedding Multi-Fuel Combustion"

_applsci, doi:10.3390/app12083951_

Round 1

Reviewer 1 Report

The manuscript deals with the development and validation of a two zone model to predict fire development in a compartment where multi-fuel combustion occurs. The model results authors compared to the experimental data and fire scenarios were very well reproduced by the model. The scientific quality of the paper is very good and the topic is relevant to the journal Applied Sciences. The manuscript is in general vey well written and it is important to highlight the difference and new insights in the present work.

Reviewer 2 Report

Comments on the revisited zone model paper.

It is nice to see a paper where simple component physical models are used to construct a zone model.

The paper is well written and it has the need for the authors to explain how they derive the “Novel” momentum Eq.(9). The application using a surrogate fuel is welcome. The rest of the paper is not dissimilar to CFAST. It would be nice alos to compare the paper with Japanese model by Tanaka san.

My comments are the following:

  1. It is not clear what is control volume for Eq.(9). What are the values of pu and pl and where are they defined in the smoke layer in Eq.(9). Do they have the same meaning as in Eq. (14)?
  2. It seems that Eq. (9) needs careful derivation. What are the forces pushing the smoke layer downwards?
  3. The entrainment Eq.(12) is not correct . In the flaming region the entrainment should be proportional to z5/2 NOT to z.
  4. The calculation of the species in the upper layer should be done using the mixture fraction conservation equation.

It is remarkable that the predictions agree well with the results and this needs some discussion given the many properties and variables used.

Reviewer 3 Report

This paper presents a zone fire model embedding  multi-fuel combustion and also validates this model. Overall it is an interesting study and can be useful for the combustion community. I recommend the publication of the paper if the authors can address the following suggestions.

1) The figures in the whole paper quality are not high and clear. This needs to be improved a lot.

2) The introduction part is too short and also the references are not adequate. ONLY 16 references were given in the paper and this could not be enough for the international journal.

3) How did the authors considered the combustion mechanism in the model? Is there any detailed reaction mechanism considered?

Round 2

Reviewer 1 Report

I recommend accepting the article in present form.

Author Response

We thank the reviewer for his careful reading of the paper and his positive and constructive comments.

Reviewer 2 Report

I would suggest that Eq. 9 still needs the definition of the control volume which should start from the ceiling extending to the smoke interface. Then all forces can be properly identified.

In addition , the direction of the positive and negative momentum and forces should be carefully accounted for in Eq. 9. The momentum of the plume is upwards whereas the momentum of the smoke mass layer goes downwards.

Is there a difference in results if Eq. 9 is not used as done in CFAST?

The entrainment equation ( Eq. 12) could be improved based on recent work after 1986.

Author Response

Dear Reviewer,

The authors thank you for your careful reading of the paper and your positive and constructive comments.

Please find below the responses to your comments:

 I would suggest that Eq. 9 still needs the definition of the control volume which should start from the ceiling extending to the smoke interface. Then all forces can be properly identified. In addition , the direction of the positive and negative momentum and forces should be carefully accounted for in Eq. 9. The momentum of the plume is upwards whereas the momentum of the smoke mass layer goes downwards.

The text has been modified to address this comment as follow:

As previously mentioned, a momentum equation is added to calculate the displacement rate of the smoke layer interface . It is obtained by applying the variable-mass Newton’s second law to the control volume extending from the interface to the ceiling (i.e., the upper layer). The Newton’s law states that the sum of all forces that act upon the control volume is equal to the net rate of mechanical momentum relative to the control volume, which leads to:

Eq.(9) modified

Is there a difference in results if Eq. 9 is not used as done in CFAST?

This cannot be done without changing the system of equations. Unlike CFAST, equation (9) governs the evolution of the volumes of the upper and lower layers according to their respective pressure. If Eq. (9) is not used, it would be necessary to reformulate the model by deriving an equation for the total pressure in the compartment and an equation for the volume of the upper layer, as done in CFAST. A comparison with CFAST could be made, but the observed discrepancies could be due not only to Eq. (9), but also to the other differences between the two models (ten-wall radiation model, convective heat transfer at the walls, taking into the ceiling jet, vent flows, numerical resolution, etc.).

The entrainment equation ( Eq. 12) could be improved based on recent work after 1986.

Until now, the most popular zone models  and design guidelines (e.g., NFPA 92, EN 12101-5, BRE 368, etc.) use plume entrainment correlations developed by Thomas, Zukoski, McCaffrey, Delichatsios or Heskestad at the end of the 1970s  and the 1980s (e.g., Heskestad and McCaffrey in CFAST; Heskestad, McCaffrey, Thomas, and Zukoski in Ozone; or McCaffrey and Delichatsisos in BRANZFIRE, among others). The correlation of Heskestad is used in the present study. However, it would be possible to evaluate a more recent (reliable) correlation, but which one ?